# BOWTIE-FLOW: EFFICIENT HIGH-RESOLUTION VIDEO GENERATION WITH PRIOR PRESERVATION

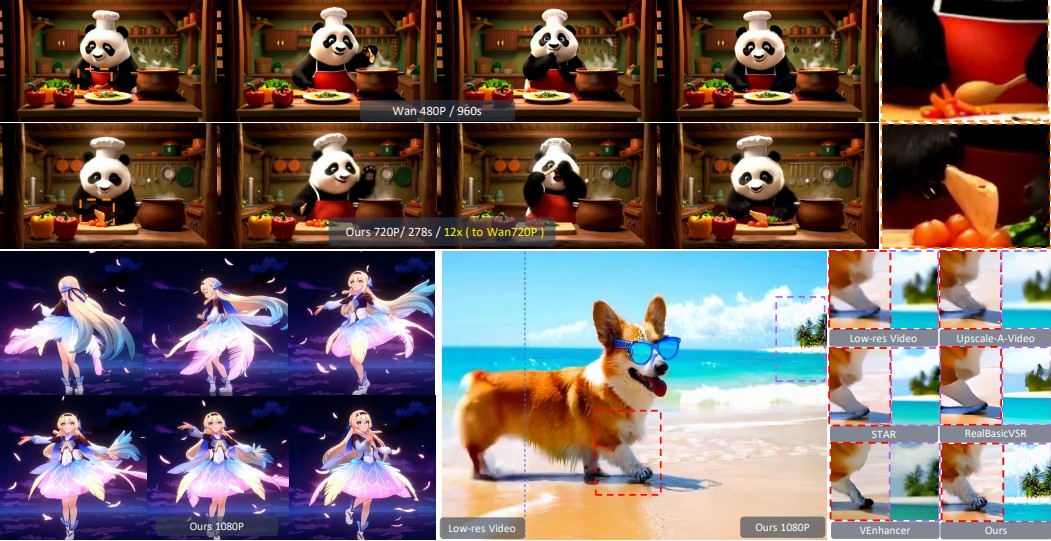

Figure 1: **Bowtie-flow generates high-resolution and high-quality videos efficiently**. It acts as a simple plug-in and brings more than 12x speed-up. At the same time, it preserves the prior (*i.e.,* layout, semantic, motion, *etc.*) of the baseline model (as the first two rows).

## ABSTRACT

The demand for high-resolution video generation is growing rapidly. However, the generation resolution is severely constrained by slow inference speeds. For instance, Wan 2.1 require over 50 minutes to generate a single 720p video. While previous works explore accelerating video generation from various aspects, most of them compromise the distinctive `priors` (*e.g.,* layout, semantic, motion) of the original model. In this work, we propose `Bowtie-flow`, an efficient framework for generating high-resolution videos, while maximally keeping the pretrained `priors`. Specifically, `Bowtie-flow` divides video generation into two stages: First, we leverage the pretrained model to generate a low-resolution preview in fast speed; then we deisign a Refiner to upscale the preview. In the preview stage, we identify that directly inferring a model (trained with higher resolution) on lower resolution causes severe prior losses. So we introduce noise reshifting, a training-free technique that mitigates this issue by conducting initial denoising steps on the original resolution and switching to low resolution in later steps. In the refine stage, we establish a mapping relationship between the preview and the high-resolution target, which significantly reduces the denoising steps. We further integrate shifting windows and carefull design the training paradim to get a powerful and effcient Refiner. In this way, `Bowtie-flow` enables generating high-resolution videos efficiently while maximally closer to the `priors` of the given pretrained model. `Bowtie-flow` is conceptually simple and could serve as a plug-in that is compatible with various basemodel and acceleration methods. For example, it achieves 12.5x speedup for generating 5-second, 16fps, 720p videos.

A bearded man **lifts a frosty pint glass** filled with amber. He **takes a slow, appreciative sip,** his eyes closing momentarily.

A high-speed video of **a splash** created by **a stone thrown into a pond.**

Figure 2: **Demonstrations for prior preservation.** The distilled model loses the prior of the base-model, which could bring performance t, our method maintains the layout, semantics, and motion, thus achieving acceleration with no loss of fidelity.

# 1 INTRODUCTION

Video generation (Yang et al., 2024; Kuaishou, 2024; Wang et al., 2025a) has witnessed remarkable advancements in recent years, with sophisticated models continually pushing the boundaries of quality and capability. However, the computational cost remains a critical bottleneck, significantly hindering further advances towards higher resolution and richer details. For example, it costs roughly 50 min to generate a 5s 720p video for the current SoTA video generation models (Wang et al., 2025a; Kong et al., 2024) and still 6 min for distillation model (Zhang et al., 2025a).

Facing this challenge, existing researches explored various strategies to make the generation process more efficient. Specifically, methods (Luo et al., 2023; Starodubcev et al., 2025) leverage step distillation to reduce the total denoising steps. Studies like (Zhang et al., 2025b; Ding et al., 2025) primarily focus on attention sparsity to improve efficiency. Furthermore, cascade multi-scale generation techniques (Ren et al., 2024a; Chen et al., 2025) have been proposed to enhance the efficiency of high-resolution image generation.

Although these methods could improve the generation efficiency, most of them inevitably compromise the intrinsic priors of the original model as shown in Fig. 2. The model's prior is reflected in its preferences for aesthetic style, semantically aligned layout and motion dynamics, *etc*. Preserving these priors is significant when accelerating a given model, as they act as a signature of the model and could directly reflect the generation quality.

To achieve this goal, we propose an efficient framework for high-resolution video generation, called `Bowtie-flow`. This framework divides the whole process into two stages based on their natural features. In **Preview stage**, we leverage a powerful pretrained model to generate low-resolution previews. The pre-k steps focus on high noise period which determines the main content of the entire video and the post-k steps uses the low resolution latent to gain speed. We analyze that each pretrained model has its own optimal resolution (usually the training resolution). While distilling the model could increase speed in a large, it often can not access the large batch of pretraining data and would compromise the original priors. Inspired by the observation that the early denoising steps determine the overall content, and the later steps refine the details, we introduce noise reshift method to address this problem. Specifically, we start from the pretrained model's optimal resolution in the early denoising steps, then switch to a lower resolution for the remaining steps. In this way, the low-resolution preview could be efficiently generated and preserve the priors (layout, semantic, motion, *etc.*) of the pretrained model. In **Refine stage**, we train a powerful and efficient Refiner through establishing a mapping relationship between the low-resolution preview and the high-resolution target. It significantly reduces the NFE(number-of-evaluations) to 10 while enriching details and correcting unreasonable artifacts. In addition, we also integrate the shift window method into the upscaling model to further reduce the computation.

As shown in Fig. 1, our method generates high-detailed and high-quality videos with priors closer to the base model (Wan 2.1) and achieves a huge speedup. Besides, our `Bowtie-flow` paradigm allows users to generate multiple previews efficiently at the same time and pick the satisfied content for further refinement to arbitrary size. Experiments show that `Bowtie-flow` achieves 12.5× speedup compared with Wan 2.1 14B for generating 5-second, 16fps, 720p videos and 8.7× speed up in HunyuanVideo 13B with 24fps.

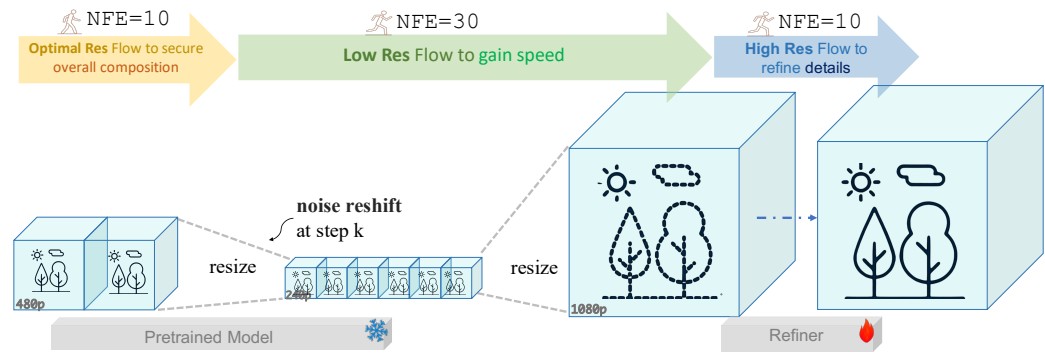

Figure 3: **An overview of the `Bowtie-flow`.** Our framework employs a powerful pretrained model (e.g., Wan 2.1) and a lightweight model which we call Refiner. Together, they execute the denoising process through an OptimRes-LowRes-HighRes latent flow (`Bowtie-flow`), ensuring a comprehensive generation from coarse semantics to fine details.

## 2 RELATED WORK

**High-resolution image and video generation** approaches can be broadly categorized into training-based and training-free. Training-based methods combine architectural innovations and model fine-tuning strategies. Turbo2K (Ren et al., 2025) accelerates 2K video synthesis by leveraging a compressed latent space and knowledge distillation within a hierarchical two-stage framework, ensuring structural coherence. UltraPixel (Ren et al., 2024b) generates 4K-resolution images using cascade diffusion models that feature shared parameters and scale-aware layers, minimizing additional parameters for high-resolution outputs. PixArt-$\Sigma$ (Chen et al., 2024) and ResAdapter (Cheng et al., 2025) enhance base models through fine-tuning but remain resolution-constrained. Other approaches like ResMaster (Shi et al., 2024) and HiPrompt (Liu et al., 2024) introduce multi-modal prompting mechanisms at the expense of computational efficiency. Training-free methods adapt inference strategies or architectures without training. MultiDiffusion (Bar-Tal et al., 2023) and its variants (*e.g.*, SyncDiffusion (Lee et al., 2023), Demofusion (Du et al., 2024)) use sliding-window denoising but suffer from repetition or computational redundancy. ScaleCrafter (He et al., 2024), FouriScale (Huang et al., 2024a), and HiDiffusion (Zhang et al., 2025e) modify network structures but risk suboptimal performance. SVG (Xi et al., 2025) investigate sparsity in the attention module but get limited acceleration. Jenga (Zhang et al., 2025g) combines multiple acceleration strategies but suffers from degradation in the original quality.

**Efficient video generation** is challenged by the quadratic complexity of attention mechanisms, especially at high resolutions. Several works (Cai et al., 2023; Xie et al., 2024; Wang et al., 2020; Choromanski et al., 2020; Yu et al., 2022; Katharopoulos et al., 2020) transform attention into linear operations to reduce complexity, while others use hybrid strategies combining local and global attention to focus only on important token pairs (Xi et al., 2025; Zhang et al., 2025b;c; Xia et al., 2025). FlashAttention Dao et al. (2022) introduces a patch-divided acceleration method . More recently, Wang et al. (2025b) proposes variable-sized window near spatiotemporal boundaries to better support long video sequences. For faster generation, rectified flow models with straight ODE trajectories (Esser et al., 2024) have been proposed in text-to-image (T2I) tasks. However, applications in text-to-video (T2V) remain limited due to the added complexity of the temporal dimension. While some methods (Ding et al., 2024; Zhang et al., 2025f) reduce the number of denoising steps, they remain limited to long-frame, high-resolution videos and overlook the domain gap between different resolutions.

## 3 METHOD

In this work, we propose `Bowtie-flow`, a framework that significantly improves the efficiency of pretrained video generation models while maximally preserving their generative priors.

## 3.1 OVERALL PIPELINE.

The overall framework of `Bowtie-flow` is shown in Fig. 3. Current video acceleration methods often operate at a fixed latent scale. While strategies(Zhang et al., 2025b; Xi et al., 2025; Yang et al., 2025) that exploit the sparsity of video tokens within attention modules can reduce computational costs, their acceleration potential remains limited. Moreover, adopting an aggressive token dropping policy—even retaining specifically selected important tokens—inevitably degrades the learned generative priors. Therefore, we analyze that the key factors that influence the generation speed are the **resolution** and the **number of denoising steps**.

Under this principle, we design a bowtie-like pipeline: we first generate low-resolution previews; then we add more details to the preview with fewer denoising steps. `Bowtie-flow` introduces a dynamic scaling mechanism that allocates a variable number of tokens according to the denoising timestep. Instead of permanently discarding tokens, we resize the latent scale to modulate the token count, thereby ensuring that the global information from the entire token set is preserved.

Specifically, in the preview stage, we retain the full capacity of a powerful pretrained model to establish the global structure with content and accelerate the whole process by dynamic scaling mechanism ; In the refine stage, we switch to a lightweight model to both accelerate the process and enrich the details.

## 3.2 PREVIEW STAGE

In this stage, we aim to leverage the strong generative capabilities of a given pretrained model for generating a low-resolution preview. We expect the low-resolution preview to preserve the priors (layout, semantic, motion) of the given pretrained model. A straightforward solution is to let the pretrained model infer on the lower resolution. However, we find that each model has its own optimal resolution; inferring on mismatched resolution causes severe prior losses. Inspired by the fact that the overall structure is determined by the initial denoising steps, we introduce a progressively downsampling method.

**Noise reshifting.** During the denoising phase, we begin with the initial gaussian noise $z_1 \in R^{b \times c \times f \times h \times w}$ and progressively downsample the clean latent representation $z_0$ with reduced spatial dimensions $h' \times w'$. Here $b$, $c$, $f$ and $h, w$ denote the batch size, number of channels, number of frames, and resolution of the model's optimal generation, respectively. Our strategy is fundamentally structured around a turning point step k along the denoising trajectory and divides into pre-k steps and post-k steps.

Before reaching k(pre-k steps), the latent representation is denoised through an ODE-based flow matching approach, described by the following iterative update:

$$\mathbf{z}_0 = \mathbf{z}_1 + \int_1^0 \mathbf{u}_\theta(\mathbf{z}_t, t) \, dt \tag{1}$$

where $\mathbf{u}_\theta(\mathbf{z}, i)$ represents the model-predicted direction function. Upon reaching step $k$, we estimate the clean latent representation as $\hat{\mathbf{z}}_0 = \mathbf{z}_k - \sigma_k \cdot \mathbf{u}_\theta(\mathbf{z}_k, k)$, where $\sigma_k$ denotes the noise standard deviation at step $k$. Subsequently, we apply a spatial downscaling operation to the estimated latent, $\hat{\mathbf{z}}_0^\downarrow = \text{Downscale}(\hat{\mathbf{z}}_0)$. where the simple linear downscale operation is implemented in the latent space. This choice is crucial as it effectively reduces spatial resolution while vigilantly preserving essential spatiotemporal coherence. Afterwards, we reinject noise which is shifted to timestep $k$, into the reduced-resolution latent space, allowing the stochastic denoising process to seamlessly resume. The reshifting process is represented by:

$$\mathbf{z}_{k-1} = \hat{\mathbf{z}}_0^\downarrow + \sigma_k \cdot \tilde{\boldsymbol{\epsilon}}, \quad \tilde{\boldsymbol{\epsilon}} \sim \mathcal{N}(\mathbf{0}, \mathbf{I}). \tag{2}$$

In post-k steps, latents denoises in lower-resolution to gain speed. This multi-scale denoising framework enables the model to first gain global semantics at coarser scales, and then get the preview at finer scales.

## 3.3 REFINE STAGE

To alleviate the computational overhead, we employ a lightweight model with 1B parameters, which reduces the time cost per step (detailed architecture can be seen in Appendix. B.1).

**Flow mapping.** We start from the preview and linear upsample it to form blurred low-resolution latents. Then we establish a mapping between low-resolution latents $z_{lr}$ and high-resolution latents $z_{hr}$ by modifying the flow-matching equation 1, substituting $z_1$ with $z_{lr}$ and $z_0$ with $z_{hr}$. Our light-weight Refiner learns $z_{lr}$'s directional information, facilitating high-quality video generation in a few denoising steps. More details can be found in Appendix B.2.

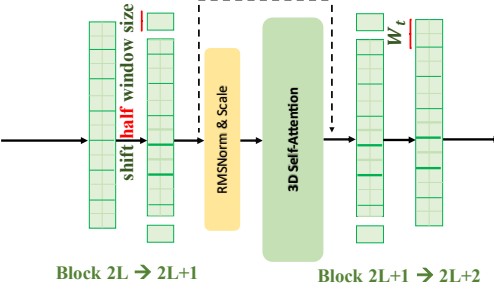

Figure 4: **Shift window** across adjacent blocks.

**Training paradigm.** We first obtain large amounts of high-quality videos and apply both pixel- and latent-level degradations to simulate low-quality video values. Pixel-level degradation simulates blur, while latent-level degradation prevents the task from becoming a trivial super-resolution problem, encouraging the model to exploit its inherent generative ability. These low-quality latents are then paired with the original high-quality latents to form the training data. However, with large resolution and long frame numbers, it presents a computational challenge and results in considerable time consumption per denoising step.

**Model structure.** Given a video feature $\mathbf{X} \in \mathbb{R}^{T \times H \times W}$, our transformer design addresses this challenge by balancing efficiency and temporal connectivity. The first transformer block applies *regular window attention* with a $t \times H \times W$ window. Specifically, $\mathbf{X}$ is divided into $\left(\frac{T}{t} + 1\right) \times H \times W$ windows, where each window spans $t$ consecutive frames along the temporal dimension while fully covering the spatial dimensions. This windowed partition allows local temporal interactions to be captured before expanding to broader temporal receptive fields in deeper blocks.

Our solution further integrates a cyclic shift-window strategy into the 3D self-attention modules shown in Fig. 4. This strategy, embedded within Transformer blocks, establishes full temporal connectivity through a two-phase cycle across layers. Each consecutive layer pair collaboratively connects all frames while maintaining computational efficiency.

In detail, Transformer block $2L$ applies 3D self-attention to non-overlapping temporal windows of size $W_t$, adjusting position embeddings for local awareness. The subsequent Transformer block $2L + 1$ applies a temporal shift of $S_t = \frac{W_t}{2}$ to the input feature, then partitions it into windows of size $W_t$ for attention computation. The cyclic-shifted attention mechanism can be expressed as:

$$\mathbf{X}^{(2L)} = \text{Attention3D}(\text{Partition}(\mathbf{X}, W_t)),$$

$$\mathbf{X}_{\text{shifted}} = \text{CyclicShift}\left(\mathbf{X}^{(2L)}, \frac{W_t}{2}\right),$$

$$\mathbf{X}^{(2L+1)} = \text{Attention3D}(\text{Partition}(\mathbf{X}_{\text{shifted}}, W_t), \text{Mask}).$$

(3)

When we shift the window by half the window size forward, one boundary window (e.g., the first window) contains temporally unrelated halves. Therefore, we apply an attention mask to separate them. We also employ position-frequency embeddings with 3D RoPE (Su et al., 2024) within each window to avoid the resolution bias introduced by fixed positional embeddings. This two-block cycle (unshifted/shifted layers) guarantees global temporal connectivity, significantly reducing memory and accelerating attention calculation for large latent tensors.

## 4 EXPERIMENTS

### 4.1 IMPLEMENTATION DETAILS

**Settings.** The Refiner of `Bowtie-flow` is trained on 24 NVIDIA A800 GPUs (80GB each) with a total batch size of 24. We finetune the transformer using the AdamW optimizer with a learning rate of 5e-5. We create a synthetic dataset of 100k LR-HR video frame pairs following the methodology in (Wang et al.). To optimize training efficiency and stabilize convergence, we employ a progressive training strategy where the model is trained at incrementally increasing frame num. We first train `Bowtie-flow` on 21-frame inputs for 1k iterations, and then extend the input length to 81 frames and finetune the model for 4k iterations.

| | | | | | | | | |
|---|---|---|---|---|---|---|---|---|
| Motion Dynamic | 41.14% | 26.43% | 32.43% | 52.13% | 20.85% | 27.03% | 44.39% | 25.00% | 30.61% |
| Video Fidelity | 46.24% | 29.73% | 24.02% | 44.40% | 23.94% | 31.66% | 74.50% | 13.76% | 11.74% |
| Text Alignment | 35.14% | 34.83% | 30.03% | 44.40% | 25.48% | 30.12% | 64.19% | 20.61% | 15.20% |
| Overall Quality | 46.24% | 18.32% | 35.44% | 55.21% | 15.06% | 29.73% | 72.64% | 13.85% | 13.51% |

■ Better ■ Same ■ Worse

Ours vs Wan2.1 — **14.05x Faster**  Ours vs SVG — **10.89x Faster**  Ours vs DMD — **1.13x Faster**

Figure 5: **User study.** We report pair-wise preference rates. `Bowtie-flow` achieves comparable quality to Wan 2.1 with a significant speedup, and outperforms SVG and DMD.

**Evaluation and metrics.** For speed assessment, we report the DiT forward pass time on NVIDIA A800, as the VAE decoding component remains constant across all configurations. We also report FLOPs to provide an intuitive comparison of computational complexity. For qualitative evaluation, we construct a video dataset of 381 low-resolution videos with prompts from VBench (Huang et al., 2024b), Videophy (Bansal et al., 2024) and PhyGenBench (Meng et al., 2024); We evaluate each prompt with a fixed seed to ensure reproducibility. Additionally, we conducted a user study to assess human preference rates between `Bowtie-flow` and various efficient generation methods. As for 1080p video generation, most existing acceleration methods are not capable of handling this setting. Therefore, for a fair comparison, we benchmark our approach against super-resolution(SR) methods and adopt several widely used video SR assessment metrics. Specifically, we employ DINO (Caron et al., 2021) and CLIP (Radford et al., 2021) to evaluate frame quality via feature similarity across frames; LAION aesthetic predictor (Schuhmann et al., 2022) to assess artistic and beauty value perceived by humans towards each video frame, and DOVER (Wu et al., 2023) to measure overall video quality.

## 4.2 COMPARISONS

We evaluate our method from both efficiency and quality perspectives. Specifically, We compare it with the Wan 2.1 baseline, which employs the FlowMatch scheduler with 50 NFE (number-of-evaluations), as well as accelerated Wan 2.1 variants utilizing 30% and 50% of the original steps. Additionally, we compare our approach with two commonly adopted acceleration techniques: the sparse attention mechanism used in SVG (Xi et al., 2025) and the bidirectional video distribution matching distillation (DMD) method (implemented following Zhang et al. (2025d)).

**Efficiency analysis.** We observe that, due to the quadratic complexity of attention, the core challenge to gain efficiency lies in reducing the number of tokens. With our proposed `Bowtie-flow`, the first stage consumes FLOPs equivalent to 480p, the second stage to 240p, and the third stage leverages a smaller-parameter model with approximately 5× fewer hidden dimensions and 2.5x fewer num heads, achieving trivial FLOPs relative to the original model. Detailed configuration is listed in Tab. R6. As shown in Tab.1, our method achieves performance comparable to Wan 2.1 at 720p resolution while reducing inference time by 12×. Furthermore, when generating 1080p videos, our approach attains a 51× acceleration compared with directly applying Wan 2.1 at 1080p.

SVG still operates on latents of the same scale and explains acceleration from the perspective of hardware-efficient tensor layout which has limited reduction in redundant tokens. DMD focuses on reducing the number of inference step. `Bowtie-flow` leverages the intrinsic properties of the denoising process by adapting latent representations at different scales to different stages of denoising, thereby achieving substantial acceleration while preserving prior information. Although the total runtime of the distillation-based method is similar to ours, we observe that DMD often fails to preserve prior information and tends to introduce unnatural color artifacts in the generated videos.

**Quality analysis.** We evaluated our model's generative performance from two perspectives: general video metrics and physics-focused assessments (as original Wan 2.1 achieves best in physical plausibility so we want to prove `Bowtie-flow` has preserved this characteristics).

As illustrated in Tab. 1, our generated video exhibits a comparable visual quality to Wan 2.1, while slightly surpassing SVG and outperforming the DMD method. We also conducted a perceptual evaluation employing a standard win-rate methodology. We have designed questionnaire with 24 randomly selected videos from the above test datasets. A total of 37 researchers in the field of video generation were asked to evaluate the results along four dimensions: Motion, Fidelity, Semantics, and Overall Quality, with the outcomes summarized in Fig. 5.

Table 1: **Quantitative comparison on Wan 2.1.** We report evaluations of the baseline (row 1), step and attention optimization methods (row 2-5) and `Bowtie-flow` (row 6). NFE=50. The highest score is in **bold** and the second highest is underlined. Abbreviations: QS (Quality Score), AQ (Aesthetic Quality), DD (Dynamic Degree), MS (Motion Smoothness), OC (Overall Consistency), SA (Semantic Adherence), PC (Physics Commonsense).

| Method | General Scene | | | | | Physical Scene | | Computation Loads | | |
|---|---|---|---|---|---|---|---|---|---|---|
| | QS↑ | AQ↑ | DD↑ | MS↑ | OC↑ | SA↑ | PC↑ | Time↓ | Speed↑ | PFLOPs↓ |
| Wan 2.1(Wang et al., 2025a) | 83.31 | 66.9 | 63.89 | 97.65 | 27.08 | 41.82 | 45.45 | 3497 (58min) | 1× | 658.46 |
| 30%step(Wang et al., 2025a) | 77.92 | 58.43 | 56.94 | 96.95 | 24.56 | 18.18 | 16.36 | 1049 | 3.34× | 197.54 |
| 50%step(Wang et al., 2025a) | 81.51 | 63.52 | 66.67 | 96.99 | 25.90 | 25.45 | 23.64 | 1748 | 2× | 329.23 |
| SVG(Xi et al., 2025) | **83.36** | 65.6 | 68.06 | 97.69 | 27.32 | 25.45 | 20.00 | 2712 | 1.29× | 429.86 |
| DMD(Zhang et al., 2025d) | 83.31 | 66.11 | 52.78 | **98.96** | 26.77 | 34.55 | 30.91 | 282 | 12.40× | 39.51 |
| Ours | 83.26 | **66.86** | **72.22** | 97.95 | **27.38** | **41.82** | **38.18** | **278** | **12.58×** | **34.3** |

The findings demonstrate that our method achieves performance comparable to the original model while surpassing many existing acceleration approaches from human perspective. From extensive examples, we observe that SVG suffers from limited robustness, as the main subjects often remain static and adds unreasonable details; while DMD tends to introduce unnatural color artifacts and produces grainy videos with reduced fidelity. We provide visualization cases in Fig. 6.

Table 2: **Plug-in integration** with other acceleration methods and different model architectures.

| Method | SA↑ | PC↑ | NFE/Time↓ | Speed↑ |
|---|---|---|---|---|
| HunyuanVideo | 29.09 | 27.27 | 50/3081 | 1× |
| +SparseAttn | 30.91 | 23.64 | 50/2775 | 1.1× |
| + `Bowtie-flow` | 43.64 | 45.45 | 50/356 | 8.7× |
| AccVideo | 32.73 | 23.64 | 5/340 | 1× |
| + `Bowtie-flow` | 36.36 | 38.18 | 5/265 | 1.3× |

Moreover, our method can be naturally extended to support 1080p video generation. Since most baseline models do not natively support this resolution, we evaluate 1080p results from the perspective of video super-resolution (SR). Specifically, we randomly select 100 samples from the aforementioned test datasets and conduct comparisons with existing SR approaches, aiming to assess the effectiveness of our method in generating high-resolution videos with enhanced detail and fidelity. We use DINO (Caron et al., 2021), CLIP (Radford et al., 2021) to assess temporal quality and LAION aesthetic predictor (Schuhmann et al., 2022), DOVER (Wu et al., 2023) to evaluate visual quality of videos. Detailed results as shown in Tab. 3. Notably, although GAN-based RealBasicVSR achieves competitive scores on some metrics, its outputs frequently exhibit excessive smoothing, do not satisfy human perceptual preferences; Diffusion-based VEnhancer also demonstrates strong generative capabilities, however, its outputs often undergo significant deviations from the input, contradicting the principle of enhancing visual quality while preserving fidelity. As shown in Fig. S13, our method demonstrates significant success in rendering details such as the paw of cat and drone shape.

**Compatiability analysis.** Furthermore, `Bowtie-flow` can function as a plug-in compatible with various diffusion model architectures and acceleration techniques. When combined with sparse attention Zhang et al. (2025g), it achieves a 7.6x speedup on HunyuanVideo (Kong et al., 2024). Additionally, it can be adapted to step-distillation models AccVideo (Zhang et al., 2025a) and achieves a 1.3x speedup. Detailed results are presented in Tab. 2.

### 4.3 VISUALIZATION RESULTS

This section presents visualizations of our `Bowtie-flow`, providing an intuitive comparison between the preview and the final results in terms of overall layout similarity as well as refinement capability. As shown in top row of Fig. 7, our design first prioritizes an efficient retention of the powerful model's inherent priors to ensure robust structural layout, semantic alignment, and motion dynamics. For instance, the ballet dancer's movements are demonstrably smooth and natural. Furthermore, our method is adept at style-aware video synthesis, exemplified by results such as a Van Gogh-style tower and a watercolor panda. Then the well-trained Refiner refines intricated details and mitigates generation artifacts. In second row, `Bowtie-flow` further enhances visual fidelity of ex-

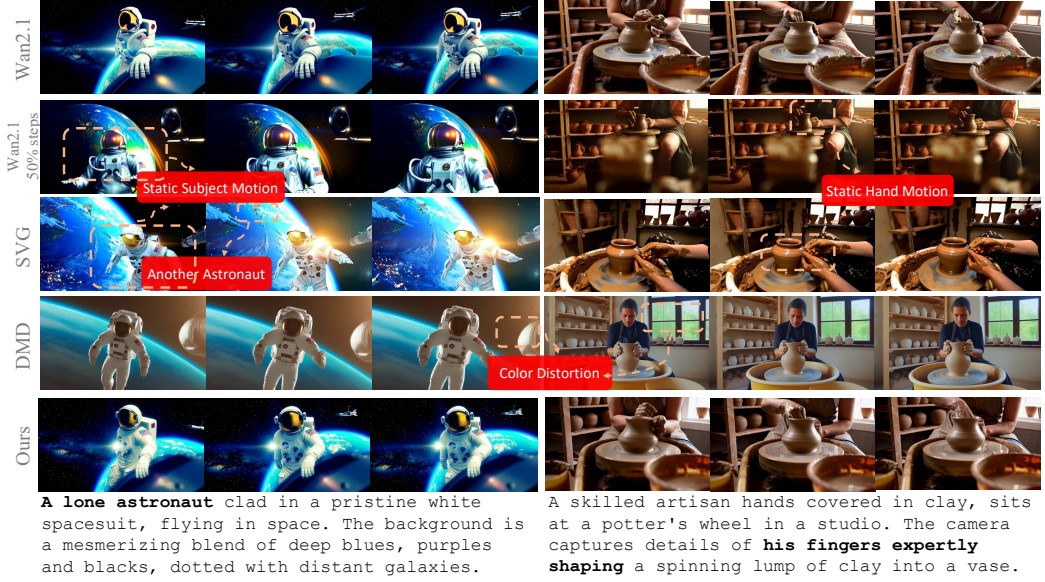

Figure 6: **Qualitative comparisons.** Our method achieves up to 12× speedup while maintaining `priors` of base model. Unreasonable contents are marked in orange. Rather than aimless camera panning, `Bowtie-flow` generates high fidelity videos with semantically aligned motion.

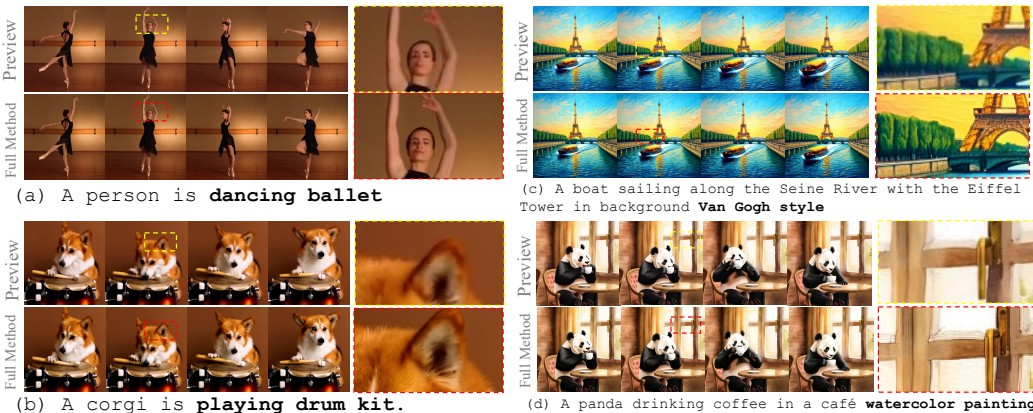

Figure 7: **Generation results for each stage**. We mark regions with artifacts and lacking detail in the preview videos using yellow boxes, while improvements from the Refiner are highlighted in red. Zoom in for a better view.

isted videos, including clearer facial and hand structures in (a), refined fur rendering on the corgi in (b), enhanced texture and structural details in (c), and more semantically consistent structural and color corrections in (d).

More examples in Fig. 1 and Appendix G further demonstrate the visual aesthetics and realism of the generated videos.

## 4.4 ABLATION

**Step division.**   In this part we look into how to balance efficiency and quality of generating a useful preview. We have found that the division of step range (i.e., the choice of k) is very important.

Since the efficiency of our method is directly proportional to the number of denoising steps performed at the initial resolution, identifying the optimal step range for the transition is crucial for balancing quality and speed. With the results presented in Tab. 4, it reveals a distinct trade-off. Employing an early transition (e.g., after step 5) accelerates the process but leads to a degradation in both layout integrity and motion quality. Conversely, delaying the transition (e.g., after step 35) not

Table 3: **Quantitative comparison** with video super-resolution method (1080p).

| Method | DINO↑ | CLIP↑ | LAION↑ | DOVER↑ | NFE/Time↓ |
|---|---|---|---|---|---|
| RealBasicVSR | 93.40 | 94.83 | 61.07 | 80.25 | 1/162.1s |
| Upscale-a-Video | 93.47 | 95.71 | 60.93 | 71.40 | 30/2517.7s |
| VEnhancer | 93.55 | 96.02 | 63.46 | 79.78 | 15/2467.6s |
| STAR | 93.68 | **96.59** | 60.81 | 63.64 | 14/912.7s |
| Ours | **93.75** | 96.30 | **63.50** | **81.20** | 10/76.5s |

Table 4: **Ablation study** of step division in preview stage. Recommend setting is in gray.

| Setting | QS↑ | AQ↑ | DD↑ | MS↑ | OC↑ | Time↓ |
|---|---|---|---|---|---|---|
| 5-35 | 82.21 | 63.45 | 66.67 | **98.23** | 27.12 | 201s |
| 10-30 | 82.01 | 62.87 | **70.83** | 98.16 | 27.51 | 252s |
| 20-20 | 81.19 | 62.54 | 69.44 | 98.05 | **27.57** | 369s |
| 30-10 | 80.78 | 61.37 | **70.83** | 98.05 | 27.52 | 481s |
| 40-0 | **82.89** | **66.57** | 68.06 | 97.70 | 27.35 | 610s |

Table 5: **Ablation study** of denoising steps in refine stage. We report `Bowtie-flow` in 720p (left block) and 1080p (right block).

| Step | 720p | | | | | | 1080p | | | |
|---|---|---|---|---|---|---|---|---|---|---|
| | QS↑ | AQ↑ | DD↑ | MS↑ | OC↑ | Time↓ | DINO↑ | CLIP↑ | LAION↑ | DOVER↑ |
| 8 | 83.24 | 66.90 | 72.22 | 97.94 | 27.34 | 244.5 | 93.70 | 96.28 | 63.48 | 80.52 |
| 9 | 83.17 | 66.90 | 70.83 | 97.95 | 27.38 | 247 | 93.75 | 96.31 | 63.48 | 80.89 |
| 10 | 83.26 | 66.86 | 72.22 | 97.95 | 27.38 | 249 | 93.75 | 96.30 | 63.54 | 81.20 |
| 11 | 83.22 | 67.03 | 70.83 | 97.95 | 27.38 | 251.8 | 93.71 | 96.32 | 63.60 | 81.57 |
| 12 | 83.31 | 66.69 | 72.22 | 97.97 | 27.37 | 254.2 | 93.73 | 96.27 | 63.49 | 81.43 |

only increases the inference time but can also compromise the final layout, as the late-stage resolution shift may disrupt an already well-defined structure. Besides, We observe from Fig. S11 that the overall compositional layout of the generated content stabilizes after approximately step 10. Based on this analysis, we identify the 10-30 step range as the optimal configuration. This range effectively preserves the structural and motion quality of the generated video while maximizing computational efficiency.

**Shift window attention.** We further investigate the impact of incorporating shift window attention in Refiner on video generation quality. As shown in Fig. 8, we observe that previously unclear and distorted hands now exhibit clear fingernail contours and appropriate wrinkles, regardless of whether the shift-window mechanism is applied. Additionally, the clarity of distant trees is noticeably enhanced. our findings suggest that a full receptive field from global attention is not critical for the refinement stage. The visual differences are negligible, indicating that local context modeling is sufficient for enhancing details at this stage.

**Inference hyperparameters.** We further investigate the influence of common inference-time hyperparameters on video generation performance. The performance trends across different steps in Refiner are visualized, with corresponding quantitative results summarized in Tab. 5. We analyze the effects of varying the number of diffusion steps and highlight the best-performing configurations in gray.

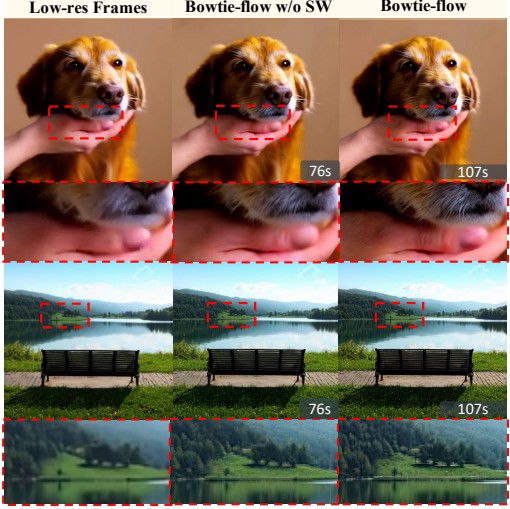

Figure 8: **Qualitative results for ablation.** SW denotes shift-window attention. The Refiner runtime on 1080p video is shown in the bottom-right corner. Our method achieves shorter runtime while preserving finer details.

## 5 CONCLUSION

We present `Bowtie-flow`, a simple yet effective framework that efficiently generates high-resolution videos while preserving the strong priors (e.g., layout, semantics, motion) of a given pretrained model. `Bowtie-flow` achieves significant improvements in both human perceptual preferences and quantitative metrics, delivering a 12x speedup for generating 5-second, 16fps, 720p videos and 51x speed up for 1080p videos in high-quality.

ETHICS STATEMENT

`Bowtie-flow` adheres to high ethical standards in machine learning and computer vision, ensuring responsible use of generative models in video and image synthesis.

REPRODUCIBILITY STATEMENT

To ensure reproducibility, we provide comprehensive implementation details, including models, datasets, and training setups. All codes on the open-source models will be released.

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

APPENDIX

**The supplementary material is organized as follows:**

## A    PROJECT WEBSITE

We have provided a local demo in the supplementary file. You can directly click on the index.html to view the videos.

Feel free to visit and explore!

## B    IMPLEMENTATION DETAILS

### B.1    BASE MODELS CONFIGURATION

In this section, we describe the configures of our baseline models. We adopt the original model implementations whenever possible.

Wan2.1 (Wang et al., 2025a): A 14-billion-parameter open-source video generation model. Wan2.1 ranks high in Physical Plausibility, ID Consistency, Scene Generation Quality etc. However with high-parameter and quadratic attention calculations for high-resolution (720p or 1080p), it suffers from large computation consume. It roughly uses over 50min to generate one 81frame 720p video on A800 80G with FA2 (Dao, 2023).

HunyuanVideo (Kong et al., 2024): A 13-billion-parameter open-source video generation model. HunyuanVideo is recognized for its smooth motion synthesis, precise semantic alignment, and high-quality aesthetics It roughly uses over 50min to generate one 129frame 720p video on A800 80G with FA2 (Dao, 2023).

AccVideo (Zhang et al., 2025a) introduces an efficient distillation method to accelerate video diffusion models using synthetic datasets. This method is adaptable to both Wan2.1 and HunyuanVideo models. In this work, we utilize the model based on HunyuanT2V. It takes approximately 6 minutes to generate a 129-frame, 720p video on an A800 80G GPU using FA2(Dao, 2023).

Upscaler: We use an internal 1-billion-parameter transformer-based latent diffusion model (Peebles & Xie, 2023) as the base T2V generation model, as illustrated in Fig. S9. We employ a 3D-VAE to transform videos from the pixel space to a latent space, upon which we construct a transformer-based video diffusion model. Unlike previous models that rely on UNets or transformers, which typically incorporate an additional 1D temporal attention module for video generation, such spatially-temporally separated designs do not yield optimal results. We replace the 1D temporal attention

Table R6: Configurations for different stages. A dash ($-$) indicates that the stage is *training-free*.

| Configuration | Preview Stage | | Refine Stage |
|---|---|---|---|
| | **Pre-$k$ Steps** | **Post-$k$ Steps** | |
| Model Para | 14B | 14B | 14B |
| Dimension | 5120 | 5120 | 1152 |
| num_heads | 40 | 40 | 16 |
| Optimizer | - | - | AdamW |
| Learning rate | - | - | $5e^{-5}$ |
| Numerical precision | bfloat16 | bfloat16 | bfloat16 |
| Resolution | 480p | 240p | 720p |
| timestep_shift | 3 | 3 | 1 |
| CFG | 7.5 | 5 | 6 |

with 3D self-attention, enabling the model to effectively perceive and process spatiotemporal tokens, thereby achieving a high-quality and coherent video generation model. Specifically, before each attention or feed-forward network (FFN) module, we map the timestep to a scale, thereby applying RMSNorm to the spatiotemporal tokens. It is worth noting that the origin model does not support video generation at 720p resolution.

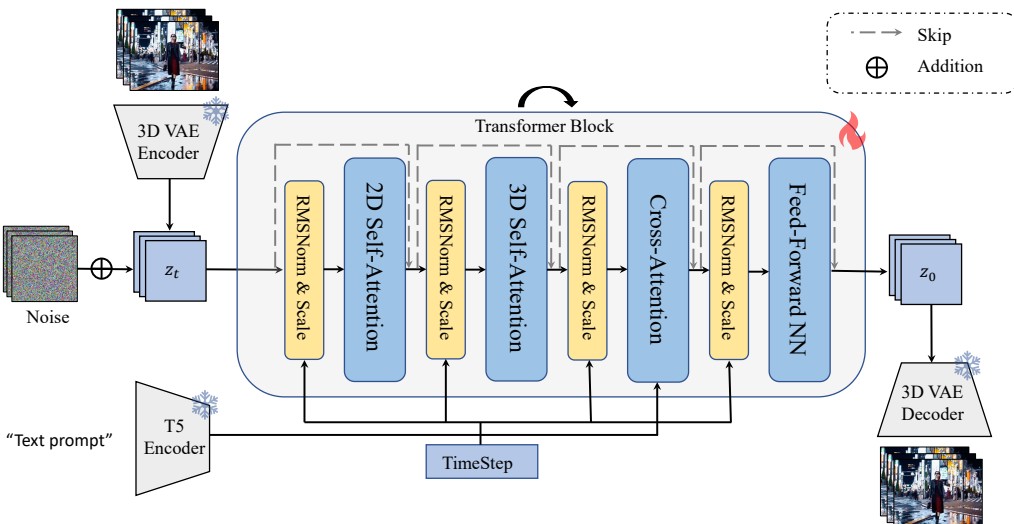

Figure S9: **Overview of the base text-to-video generation model.**

## B.2 TRAINING DETAILS

We train the upscaler to transform a blurred video into a clear one while simultaneously refining unreasonable details. Specifically, we adopt flow matching (Esser et al., 2024) to map the low-resolution latent representation $Z_{LR}$ to the high-resolution latent representation $Z_{HR}$.

Intermediate points are obtained via linear interpolation between $Z_{LR}$ and $Z_{HR}$. The training loss is defined with the target $Z_{LR} - Z_{HR}$. We randomly sample $t \sim \text{LogitNorm}[0, 1]$ with a timestep shift of 5, and compute:

$$Z_t = (1 - t) \cdot Z_{HR} + t \cdot Z_{LR}.$$

Using the $t$-independent target $Z_{LR} - Z_{HR}$ results in a straighter ODE trajectory, thereby enabling few-step generation.

Since we divide Bowtie-flow into a preview stage and a refine stage, we also provide the corresponding configurations for different stages, as summarized in Tab. R6.

Table R7: **Resolution settings across different aspect ratios.** We report the spatial dimensions (height, width) for 1080p, 720p, 480p, and 240p under square (1:1) and portrait (9:16) aspect ratios.

| Aspect Ratio | 1080p | 720p | 480 | 240p |
|---|---|---|---|---|
| 1:1 | (1440, 1440) | (960, 960) | (576, 576) | (336, 336) |
| 9:16 | (1080, 1920) | (720, 1280) | (480, 832) | (240, 416) |

## B.3 DATASET CONSTRUCTION

We collect approximately 100K high-quality video clips from the Internet to construct our training dataset. Given the highly variable quality of online videos, we follow the automated filtering pipeline proposed in (Xie et al., 2025) to retain visually high-quality content.

Specifically, we first discard videos that are overly bright or overly dark. For each remaining video, we uniformly sample 10 frames and compute two metrics: the average MUSIQ score (Ke et al., 2021) and the Laplacian variance, which reflects the level of spatial detail and sharpness. Videos with an average MUSIQ score below 40 or a Laplacian variance below 30 are discarded.

To simulate degradations, we apply both pixel-level and latent-level operations. At the pixel level, we follow (Wang et al.) to synthesize corresponding LR–HR video pairs. At the latent level, we inject noise sampled from the range $[0.6, 0.9]$.

We train `Bowtie-flow` on resolutions 720p and 1080p, while fixing the target FPS to 16 via frame skipping. For this multi-resolution training, we adopt aspect-ratio bucketing with a minimum unit size of 32 pixels. Since `Bowtie-flow` focuses on dynamically selecting suitable resolutions, frequent scale changes occur during inference and training. For clarity, we list the resolution buckets in Tab. R7.

## B.4 EVALUATION METRICS

For text-to-video evaluation, we randomly select 381 prompts, consisting of 326 prompts from the benchmark VBench (Huang et al., 2024b), 20 prompts from Videophy (Bansal et al., 2024), and 35 prompts from PhyGenBench (Meng et al., 2024). Our evaluation protocol measures video quality from both global and local perspectives. To this end, we employ a comprehensive suite of automated metrics: *Quality Score (QS), Aesthetic Quality (AQ), Dynamic Degree (DD), Motion Smoothness (MS), Overall Consistency (OC)* for general video evaluation, and *Semantic Adherence (SA)* and *Physics Commonsense (PC)* for physical plausibility.

**Quality Score (QS).** The weighted average of multiple dimensions, including subject consistency, background consistency, temporal flickering, motion smoothness, aesthetic quality, imaging quality, and dynamic degree.

**Aesthetic Quality (AQ).** Assesses the artistic and aesthetic value of each frame using the LAION aesthetic predictor (Schuhmann et al., 2022). It reflects high-level properties such as composition, color harmony, and photorealism.

**Dynamic Degree (DD).** Quantifies the magnitude of motion using optical flow fields estimated by RAFT (Teed & Deng, 2020). This metric discourages static or near-static generations and promotes natural motion in dynamic scenes.

**Motion Smoothness (MS).** Measures the temporal smoothness of motion using a video frame interpolation model (Li et al., 2023).

**Overall Consistency (OC).** Computed by ViCLIP (Wang et al., 2023) on general text prompts, reflecting both semantic and stylistic consistency.

**Semantic Adherence (SA).** Measures the alignment between the generated video and the input text prompt (Bansal et al., 2024). SA = 1 indicates that the caption is well grounded in the generated video.

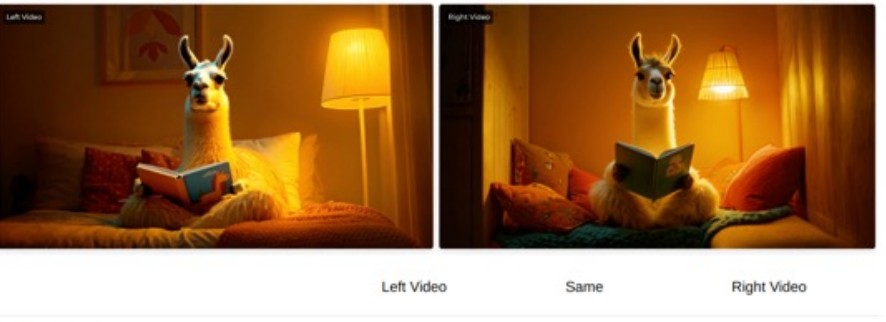

Figure S10: **User study** questionnaire form example.

**Physics Commonsense (PC).** Assesses whether the generated video intuitively follows real-world physical laws. PC = 1 indicates that object dynamics, interactions, and motions align with human commonsense physics. For evaluation, we consider PC and SA return values greater than or equal to 0.5 as PC = 1 and SA = 1, and values less than 0.5 as PC = 0 and SA = 0.

**Evaluation of 1080p Videos.** We further evaluate 1080p videos from the perspective of super-resolution quality. Specifically, we calculate the DINO (Caron et al., 2021) feature similarity across frames, evaluate the temporal consistency of the background scenes by calculating CLIP (Radford et al., 2021) feature similarity across frames, use the LAION aesthetic predictor (Schuhmann et al., 2022) to measure aesthetic quality, and DOVER (Wu et al., 2023) to assess overall video quality.

### B.5 DETAILS OF USER STUDY

Fig. S10 shows the Google Form questionnaire used in our user study to present video assets. We randomly sampled 24 prompts and constructed different comparison pairs with randomized orders to avoid positional bias. To ensure data reliability, we filtered out invalid responses in which participants consistently selected the same option across all four questions (e.g., always choosing "left video" or "same"). After this filtering, a total of 37 valid questionnaires remained, and the corresponding results are reported in Fig. 5.

## C ADDITIONAL EXPERIMENTAL RESULTS

### C.1 COMPARISONS

In this section, we first compare the preview (240p) generated by Bowtie-flow with the same resolution generated by Wan2.1. We conducted the user study using a standard win-rate methodology.

Table R8: **User preference comparison** between `Bowtie-flow`'s preview and Wan2.1 240p.

| Metrics | Better | Same | Worse |
|---|---|---|---|
| Motion Dynamics | 76.17 | 19.5 | 4.33 |
| Text Alignment | 83.37 | 13.96 | 2.67 |
| Overall Quality | 82.5 | 8.28 | 9.22 |

Table R9: **User preference comparison** between `Bowtie-flow` and FlashVideo.

| Metrics | Better | Same | Worse |
|---|---|---|---|
| Motion Dynamics | 67.72 | 24.91 | 7.37 |
| Video Fidelity | 80.43 | 16.02 | 3.55 |
| Text Alignment | 69.03 | 24.35 | 6.61 |
| Overall Quality | 89.24 | 5.27 | 5.49 |

Table R10: **Comparison** between `Bowtie-flow` and FlashVideo. Highest value in **bold**.

| Method | QS ↑ | AQ ↑ | DD ↑ | MS ↑ | OC ↑ |
|---|---|---|---|---|---|
| Flashvideo | 82.99 | 62.55 | 63.47 | 96.84 | **27.65** |
| Bowtie-flow | **83.24** | **66.86** | **72.22** | 97.95 | 27.38 |

We randomly selected 30 pairs of videos between `Bowtie-flow`'s preview and Wan2.1. Participants indicated their preferences across three key dimensions: Motion Dynamics, Text Alignment, and Overall Quality. A total of 60 completed feedback forms were collected. We report the percentage of each option. The results presented in Tab. R8 demonstrate that our method is significantly preferred over the competing method. This outcome aligns with our experimental design, confirming that OptimRes-LowRes flow preserves superior spatial layout and maintains greater semantic alignment.

We also compare `Bowtie-flow` with cascaded video generation, using FlashVideo (Zhang et al., 2025f) as an example. As shown in Fig. S12, our method generates more complex motions, achieves better text alignment, and offers enhanced visual refinement, resulting in higher overall visual quality. Quantitative results are shown in Tab. R10.

Additionally, we conducted a user study with expanded evaluation criteria. We introduced "Video Fidelity" dimension, specifically focusing on visual noise and artifacts in the generated videos. As shown in Tab. R9, `Bowtie-flow` outperforms FlashVideo in terms of user preference.

As for the Refine stage, we provide more visual results. As shown in Fig. S13, our method demonstrates significant success in rendering details such as the cat's paw and the drone's shape. We also present a sequence of four frames in Fig. S14 to further compare VEnhancer with our method. While both methods generate clear videos, `Bowtie-flow` handles dynamic details more effectively.

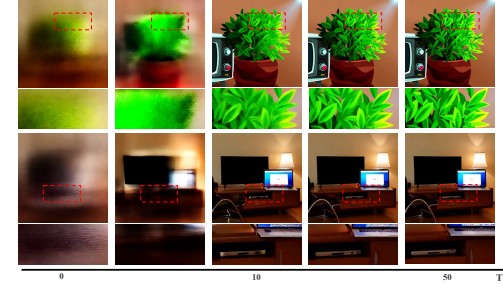

Figure S11: **Visualization of the denoising process.** The overall structure emerges rapidly within the first few denoising steps (around 10), with subsequent steps focusing on refining fine-grained details.

## C.2 ABLATION STUDIES

**Shift window attention.** We provide quantitative results in Tab. R11 to further analyze the effect of shift-window attention on video generation. Our method achieves a shorter runtime while preserving finer details, demonstrating that when handling high-resolution video, we can focus on a local receptive field to refine details.

**Step division.** Furthermore, we visualize the changes in video content during the denoising process, as shown in Fig. S11. We observe that the overall structure emerges rapidly within the first few denoising steps (around 10), with the remaining steps dedicated to progressively refining fine-grained details. Thus, we choose the denoising step to be around 10 and visualize the results of

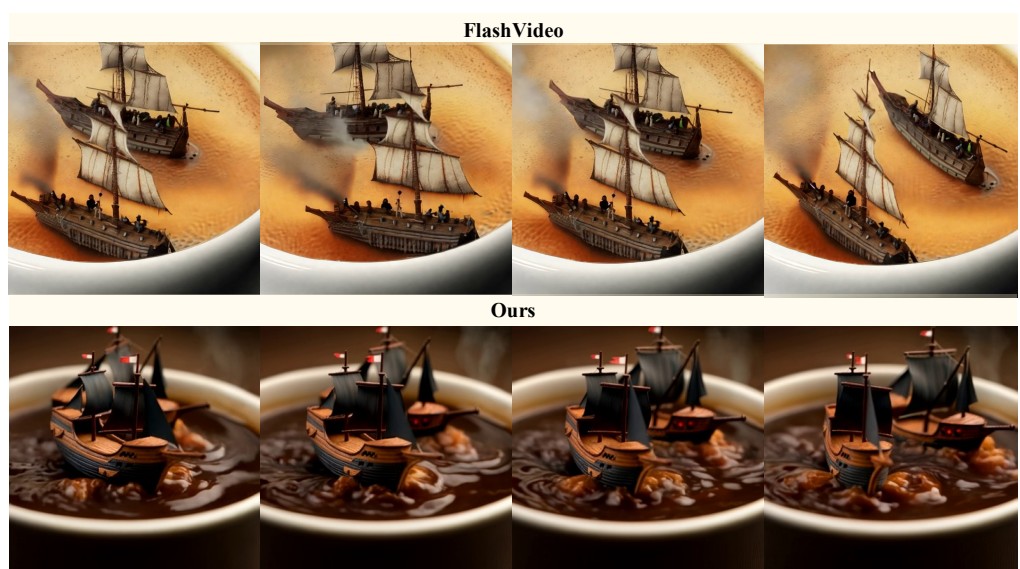

In coffee cup,tiny pirate ships battle on swirling with steamy waves.

Figure S12: **Visual Comparisons between `Bowtie-flow` and FlashVideo.** Our results exhibit a notably more imaginative and aesthetically pleasing scene of pirate ships in a coffee cup, compared to FlashVideo. The texture in our results is rendered with greater delicacy, vividly capturing the "steamy waves."

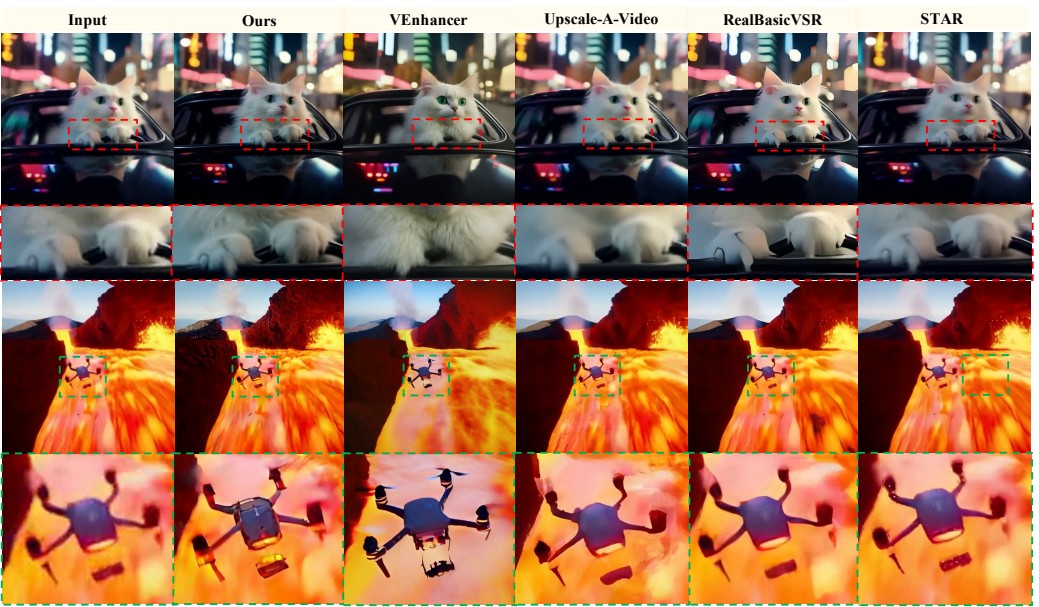

Figure S13: **Comparison with Various Video Enhancement Methods.** Our method achieves more perceptually realistic and detail-rich refinement based on the initial low-resolution previews, outperforming other approaches in consistency and visual fidelity. Specific observations for each row are annotated beneath the corresponding frames.

the refiner under different inference hyperparameters in Fig. S15. Quantitative results are shown in Tab. 4.

## D  TEASER PROMPT LIST

We provide our prompt list for the generated videos presented in Teaser 1.

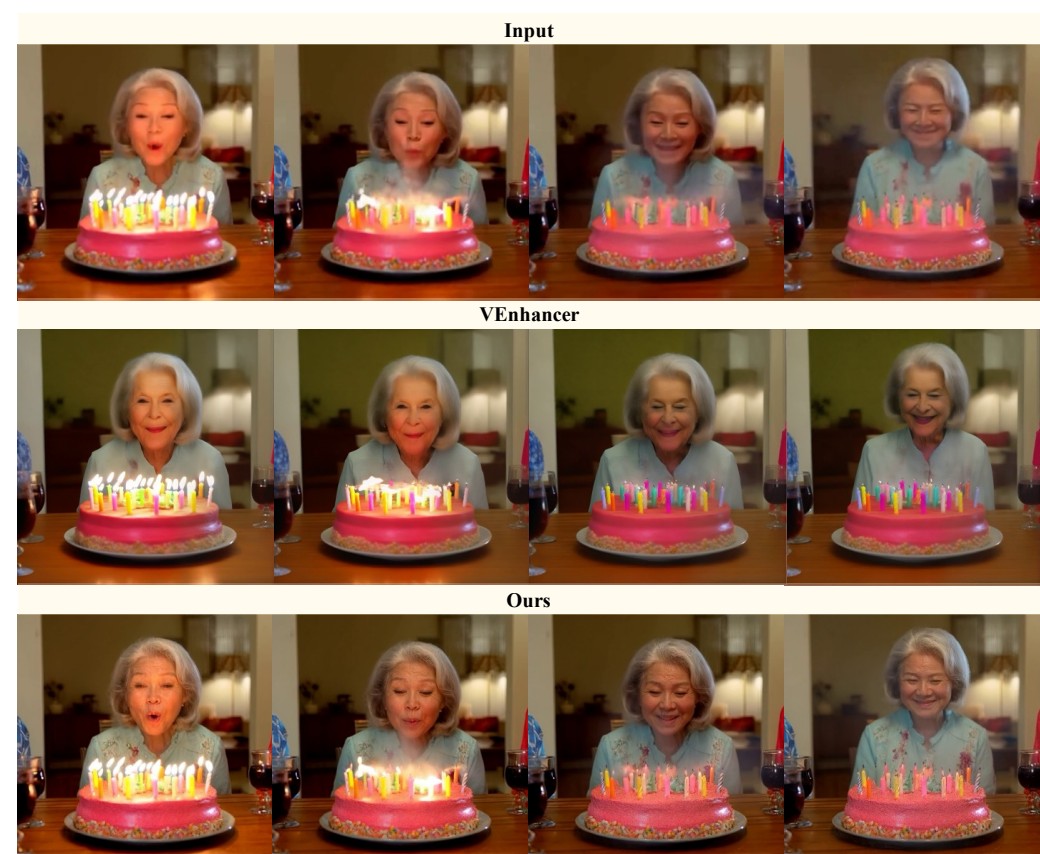

`Grandma` smiles, `blows` out `candles`—warm light and joy fill the cozy room.

Figure S14: **Comparison of fine-grained facial expressions and intricate details.** Example from a video clip with significant changes in a grandmother's facial expressions. VEnhancer struggles with facial identity, inaccurate lip articulation, and ambiguous candle flickering. In contrast, `Bowtie-flow` (Ours) intelligently refines these details, realistically augmenting them while maintaining consistency.

Table R11: **Ablation study** of key components. Highest value in **bold**.

| Method | QS ↑ | AQ ↑ | DD ↑ | MS ↑ | OC ↑ | Time↓ |
|---|---|---|---|---|---|---|
| `Bowtie-flow` _woShiftWindow | 82.94 | 66.82 | 69.44 | **98.12** | 27.41 | 107s |
| `Bowtie-flow` | **83.24** | **66.86** | **72.22** | 97.95 | 27.38 | textbf76s |

1. A charming panda, dressed in a chef's hat and red apron, chops vegetables in a rustic kitchen. It stirs a pot, tastes the soup, and plates a beautifully arranged dish, exuding delight.

2. A girl spins in the starry night sky, her shimmering pastel costume and floating feathers captured in a dreamy anime illustration.

3.A playful corgi with golden fur trots along a tropical beach, wearing blue sunglasses. The camera follows it as it walks along the shoreline, pauses, and enjoys the sun and waves.

# E    LIMITATIONS AND FUTURE WORK

Although our method is theoretically capable of generating videos at arbitrarily high resolutions, in practice it is constrained by computational resources. Specifically, our experiments show that the current implementation can stably support resolutions up to $2048 \times 2048$, while higher resolutions

Figure S15: **Results of the refiner under different inference hyperparameters.**

will lead to out-of-memory (OOM) errors on A800(80G). As part of future work, we plan to integrate patch-based spatial division strategies or memory-efficient attention mechanisms to further extend the scalability of our approach, enabling efficient training and inference at ultra-high resolutions.

## F  LLM USAGE

**Scope of use.**  We used a large language model (LLM) *only for writing polish*, including grammar correction, phrasing refinement, and improvements to clarity and readability.  The LLM did *not* contribute to research ideation, problem formulation, method design, experimental setup, result selection, interpretation, or drafting of technical content (theorems, algorithms, proofs, metrics, or analyses).  All technical claims, experiments, figures, tables, and conclusions were conceived, implemented, and verified by the authors.

## G  MORE QUALITY RESULTS

We have provided additional comparison cases, as shown in Fig.S16 and Fig.S17.

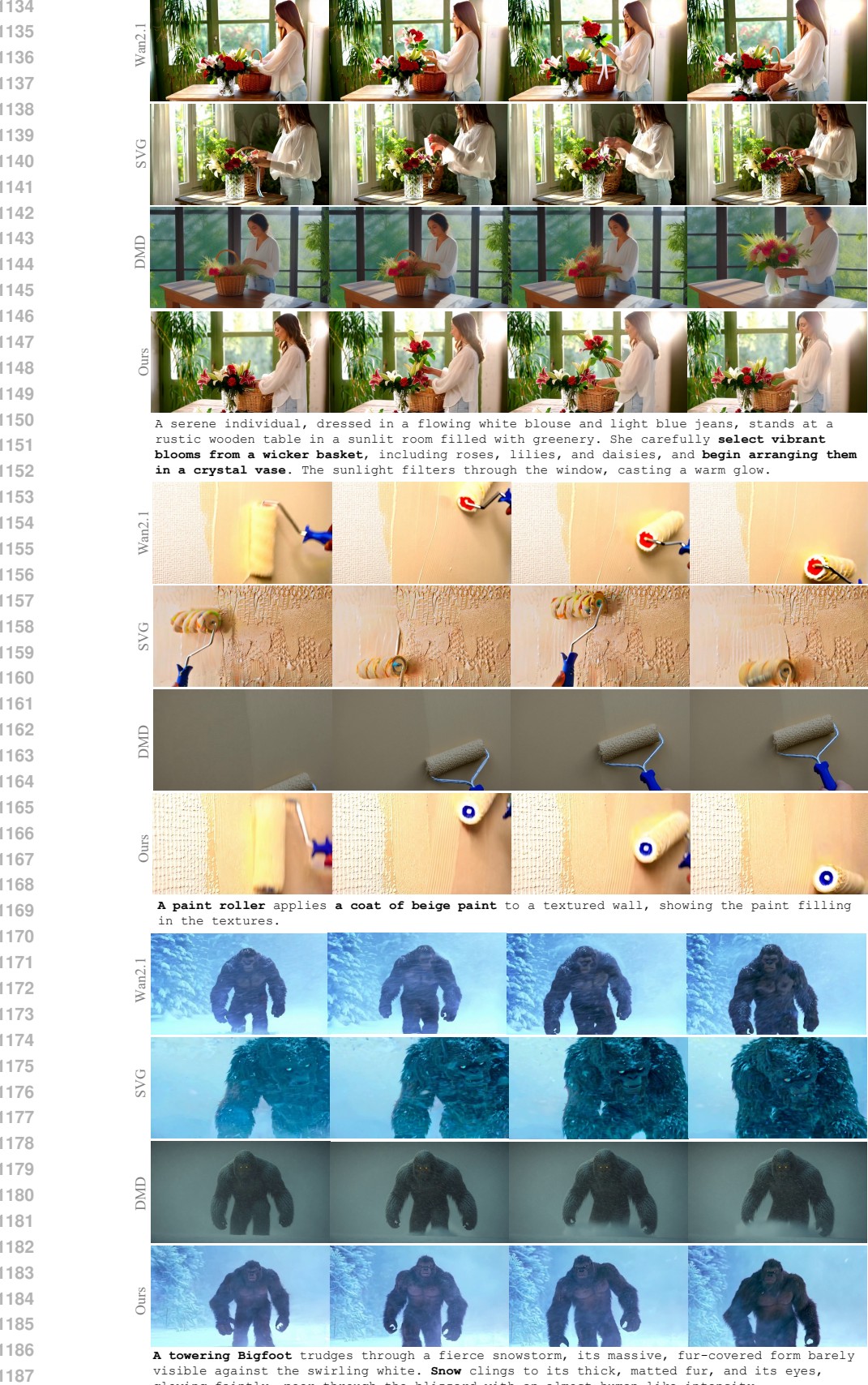

A serene individual, dressed in a flowing white blouse and light blue jeans, stands at a rustic wooden table in a sunlit room filled with greenery. She carefully **select vibrant blooms from a wicker basket**, including roses, lilies, and daisies, and **begin arranging them in a crystal vase**. The sunlight filters through the window, casting a warm glow.

**A paint roller** applies **a coat of beige paint** to a textured wall, showing the paint filling in the textures.

**A towering Bigfoot** trudges through a fierce snowstorm, its massive, fur-covered form barely visible against the swirling white. **Snow** clings to its thick, matted fur, and its eyes, glowing faintly, peer through the blizzard with an almost human-like intensity.

Figure S16: **Comparisons.** From top to bottom, each three videos is from the same setting.

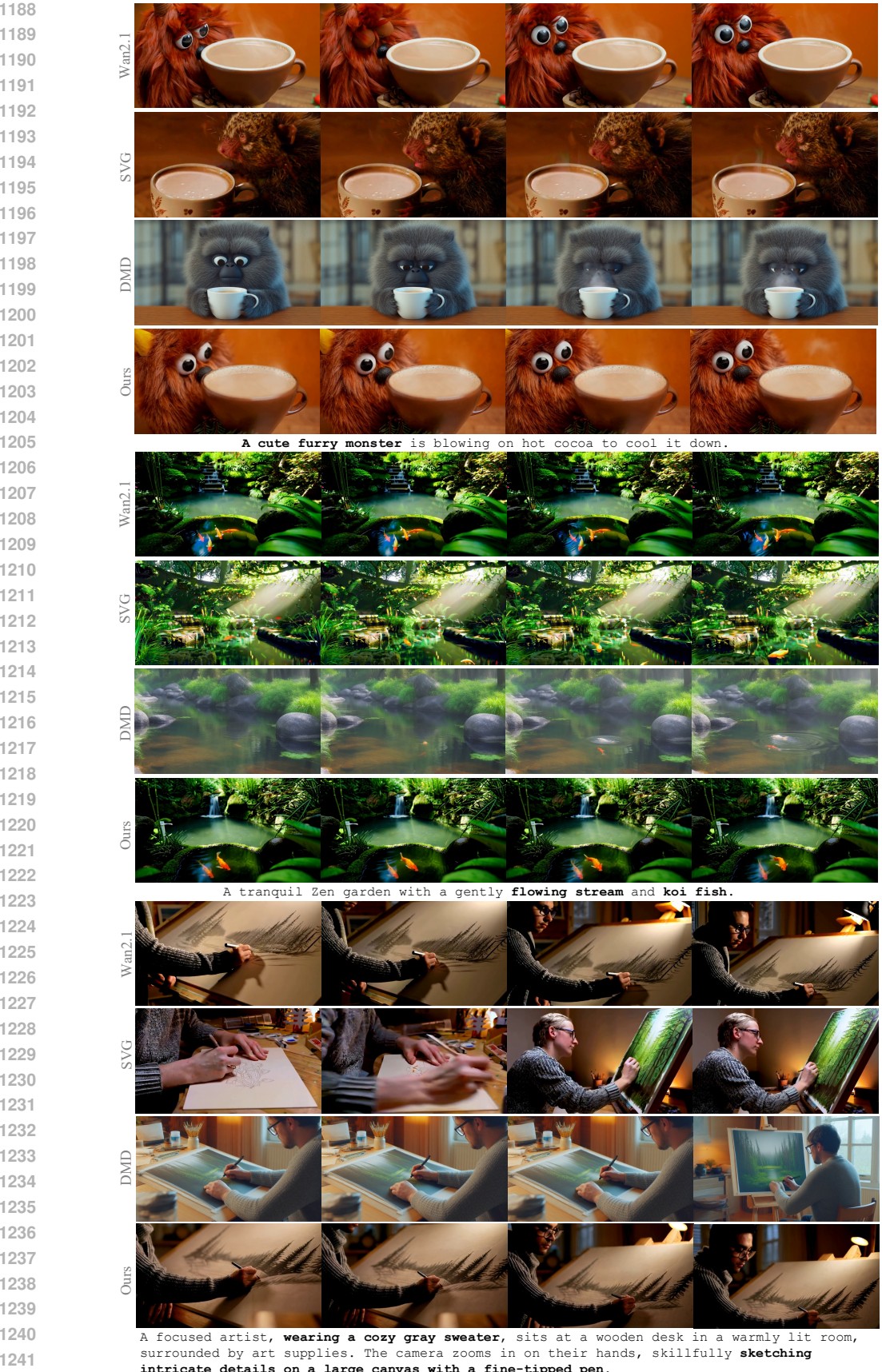

Figure S17: **Comparisons.** From top to bottom, each three videos is from the same setting.

