# OpenReview forum: "Bowtie-flow: Efficient High-Resolution Video Generation with Prior Preservation"
_ICLR.cc/2026/Conference — ICLR 2026 Conference Withdrawn Submission_

### Official Review · Reviewer_Xtqm · 2025-10-15

**Soundness:** 2
**Presentation:** 1
**Contribution:** 2
**Rating:** 4
**Confidence:** 4

**Summary:**

The authors propose to bowtie-flow to accelerate high-resolution video generation. Specifically, the method generates the first k steps in native resolution (480p), then downsample to low resolution to finish the rest (240p). Finally, the video is upsampled by a refiner model to high resolution (720p/1080p). The refiner is a lightweight network trained on paired low-res & high-res data. The method is evaluated against the original model (Wan2.1 and Hunyuan and AccVideo).

**Strengths:**

1. The model achieves speed-up while retaining quality against the baseline.

2. The method can be applied to multiple models (wan2.1, hunyuan, accvideo).

3. The paper provides standardized evaluation results and human preference evaluation.

**Weaknesses:**

1. The work's idea is highly similar to Bottleneck Sampling [1], where it also demonstrates on HunyuanVideo and Flux that the middle denoising steps can be done in low-resolution while being completely training-free. The paper should cite and compare with Bottleneck Sampling to justify the necessity of training a refiner model. I believe this is quite critical because requiring training is a big weakness compared to training-free approaches.

2. The paper has way too many grammatical problems in writing. The current paper is below the standard for acceptance.

* [1] Training-free diffusion acceleration with bottleneck sampling.

**Questions:**

1. Please address weakness 1, as I feel it is the most critical.

2. The paper claims that it has compared to DMD [1], but it actually cites VSA [2], leading to confusion. DMD is step distillation, and VSA is sparse attention without step distillation. After some research, I believe the authors are referring to FastWan [3]? If so, it should be cited as FastWan instead of DMD. Please also specify the exact NFE you are using in Table 1 and whether you are using sparse attention. Additionally, FastHunyuan can also be included as a comparison option [4].

3. Table 3 misses citations. If space is an issue, this method should be cited somewhere else. Currently, they are not in the references at all. It would also be nice to compare with recent one-step super-resolution models: SeedVR2 [6] and DOVE [7], but it is not required.

* [1] Improved Distribution Matching Distillation for Fast Image Synthesis
* [2] VSA: Faster Video Diffusion with Trainable Sparse Attention
* [3] https://huggingface.co/FastVideo/FastWan2.1-T2V-14B-Diffusers
* [4] https://huggingface.co/FastVideo/FastHunyuan
* [5] Seedvr2: One-step video restoration via diffusion adversarial post-training
* [6] DOVE: Efficient One-Step Diffusion Model for Real-World Video Super-Resolution

---

### Official Review · Reviewer_hnPU · 2025-10-16

**Soundness:** 3
**Presentation:** 2
**Contribution:** 1
**Rating:** 2
**Confidence:** 3

**Summary:**

This paper presents a training-based video diffusion acceleration method. It composes of two models: the pre-trained model and a additionally trained refiner. First, the pre-trained model operates at high resolution and then low resolution to generate the latent, where a noise reshift mechanism is introduced across different resolutions. Then, the lightweight refiner lifts the latents to higher resolution. Experiments show that the proposed framework is evaluated on wan 2.1 model and achieves significant acceleration without quality degradation.

**Strengths:**

1. The paper introduces an effective framework for accelerating video diffusion models. The multi-resolution inference strategy seems technically sound for reducing the computational cost.
2. Experiments show significant improvements in speed, while maintaining visual quality across many quantitative metrics and human evaluations.

**Weaknesses:**

1. Lack of technical contribution. The presented method (high -> low -> high resolution) largely overlaps with previous work [1], and the noise-shifting mechanism has also been studied in [2, 3]. The refiner-based approach has also been widely investigated since [4]. The authors should compare their approach with existing methods and further clarify their technical contribution.
2. Insufficient experiments. The experiments only compare against sparse attention and distillation-based diffusion acceleration methods, while ignoring existing baselines similarly using dynamic resolution strategy [1, 3]. These training-free acceleration methods should be included as baselines to better justify the claimed performance advantages and the necessity for training an additional refiner.
3. Lack of evaluation on standard benchmarks. Although the evaluation use prompts from several existing benchmarks, no quantitative results are reported on the original benchmarks themselves. It would be better for the authors to include direct results on VBench (VBench-2.0), PhyGenBench, and other commonly used benchmarks.
---
[1] Xia, et al. Training-free diffusion acceleration with bottleneck sampling. arXiv:2503.18940

[2] Jin, et al. Pyramidal flow matching for efficient video generative modeling. arXiv:2410.05954

[3] Zhang, et al. Training-free efficient video generation via dynamic token carving. arXiv:2505.16864

[4] Ho, et al. Cascaded diffusion models for high fidelity image generation. arXiv:2106.15282

**Questions:**

1. How is the refiner / upscaler trained? Are there any proposed techniques (e.g. noise augmentation) to handle the artifacts in generated low-resolution video?
2. How to ensure identity consistency with proposed framework, especially in image-to-video generation? Injecting image condition across multiple resolutions could be challenging, and the refiner may significantly change the person identity.

---

### Official Review · Reviewer_JMnw · 2025-10-24

**Soundness:** 2
**Presentation:** 2
**Contribution:** 2
**Rating:** 2
**Confidence:** 4

**Summary:**

The authors propose a method to accelerate high resolution inference for diffusion based video generation methods such as Wan 2.1 and Hunyuan Video. The authors propose performing low resolution inference for the initial steps to preserve the original model's structure, then further downsample the resolution to progress generation at a reduced cost. Finally a smaller Refiner model is trained to perform super resolution on the low resolution outputs, producing the final videos at a 12.58x speedup with respect to Wan 2.1 14B. Quantitative evaluation suggests on-par or better results with respect to the original model, but qualitative results suggest upsampling artifacts might affect the produced outputs. The supplementary material web page misses the majority of samples, impeding verification of samples quality. The paper presents some clarity issues as highlighted in the weaknesses. Some of the method components do not appear to be sound.

**Strengths:**

- The method promises a large inference time speedup of up to 12.58x for Wan 2.1 14B
- The idea of generating high resolution videos in two stages from low to high resolution, using models of different capacity to avoid incurring prohibitive quadratic self attention costs is principled, but also already explored (e.g. MovieGen)
- The idea of further speeding up the low resolution stage by decreasing resolution after the initial steps to preserve original model structure is interesting, but similar to (https://arxiv.org/pdf/2503.18940) and some concerns are present on its exact implementation

**Weaknesses:**

- The supplementary material is missing video samples. The website section misses samples showing comparison results of the method with respect to baselines. Without seeing video generation results for the proposed method it is not possible to assess its quality and check for presence of artifacts. The Fig. 1 corgi sample shows artifacts especially in the background palms and eyeglasses. Without video samples it is not possible to determine the method's quality.
- The derivation of equation in LL201 is not well introduced. I need to assume the sigma term corresponds to the timestep which is an unusual notation for flow matching.
- LL203-204 do not appear principled. Commonly used auto encoders have been shown not to possess scale equivariance (https://arxiv.org/abs/2502.14831). The choice of downsampling the latents in this stage does not appear principled in this regard. Decoding to RGB space, downsampling, and re-encoding would avoid the issue.
- Eq 2 appears incorrect if we assume a flow matching formulation with forward process defined as x_t = (1-t) x_0 + t eps. Why is z_0 not scaled according to sigma? Unclarity is emphasized by the use of sigma instead of the timestep.
- Eq 2 does not appear principled. Applying rejoicing using the same timestep to a tensor of a reduced resolution will alter its effective SNR, resulting in a tensor with a greater noise level. It is unclear if this is done intentionally or by mistake. This should be clarified in the manuscript.
- Fig 4 should ideally show two blocks to better show the window shifting
- Please describe that metrics in Table 1.1 belong to VBench and include VBench overall score.
- Many typos LL65, LL79, LL89, LL91, LL165, LL178, LL195
- Table R6 seems to report an incorrect number of parameters for the Refiner

**Questions:**

- Why does the proposed method not result in at most a tie when compared to the original Wan 2.1? It is unclear why the proposed pipeline should result in improved model performance with respect to vanilla Wan 2.1
- Could the authors please show the full scores for Table 2? Why does performance improve with respect to the baseline model which should serve as an upper bound?
- Please clarify in LL35 the model size to clarify the statement
- LL821 reports that "100k high-quality video clips from the Internet" are used. Could the authors comment on how these videos were acquired and their license?
- Authors report the DMD implementation follows "FastVideo", but details are not specified. Can the authors describe the exact implementation and hyperparameters that were used to obtain the distilled checkpoint?

---

### Official Review · Reviewer_YjRU · 2025-10-28

**Soundness:** 2
**Presentation:** 1
**Contribution:** 2
**Rating:** 4
**Confidence:** 4

**Summary:**

This paper proposes Bowtie-flow, a framework for efficient high-resolution video generation that aims to preserve the priors (layout, semantics, motion). Bowtie-flow consists of two stages: (i) Preview stage and (ii) Refiner stage. Bowtie-flow achieves acceleration by a large margin compared to previous base models.

**Strengths:**

### Motivation
- Generating a video at high-resolution is a desirable technique in video generation.

### Method
- To efficiently generate preview videos, timestep reshifting is used.
- A lightweight refiner is trained with shift-window attention mechanism.

### Experimental results
- Bowtie shows a large improvement over inference speed.
- Bowtie also achieves a comparable (or even better) quality than baselines in quantitative metrics and a user study.

### Writing/Presentation
- The paper is overall easy to follow.

**Weaknesses:**

### Motivation
- The two-stage idea of Bowtie-flow is quite similar to many cascaded video generation frameworks (e.g., Imagen-video from Google), where the first stage is to generate a low-res video and then upscale it to high-res.

### Method
- Noise reshifting is not well supported either empirically or theoretically. It might be easier to show some visualizations of noise at different time steps as a motivation for this technique.

### Experimental results
- A quality degradation (e.g., sharpness) is noticeable compared to other models. Compared to DMD, Bowtie-flow only gains 1.01x but with a much less sharper result.

### Writing/Presentation
- Many videos are missing in "compare_4_method" in the supplementary material, which makes the proposed method much less convincing.

**Questions:**

The paper is simple and interesting, but limited to poor presentation especially many missing videos in the supplementary material. I'd suggest authors to include more video results in the rebuttal.

---

### Note · Authors · 2025-11-12

**Comment:**

We would like to withdraw our paper titled Bowtie-flow for further revision.

**Withdrawal Confirmation:**

I have read and agree with the venue's withdrawal policy on behalf of myself and my co-authors.